# Tactile Pressure Sensors Calibration with the Use of High Pressure Zones

**DOI:** 10.3390/s22197290

**Published:** 2022-09-26

**Authors:** Petr Zvyagin

**Affiliations:** Laboratory of Ice Engineering Research Fundamentals, Peter the Great St. Petersburg Polytechnic University, 195251 St. Petersburg, Russia; pnzvyagin@gmail.com

**Keywords:** pressure sensing, load patch, tactile sensor, calibration, measurement, Tekscan, ice load, high pressure zones

## Abstract

A simple and cost-effective calibration procedure for piezoresistive ink tactile pressure sensors is crucial for their use in geotechnical research applications. Such a procedure should be applicable in field conditions and require a minimum amount of equipment. The paper describes a new method for calibrating tactile pressure sensors with 8-bit sensels’ output. The method is based on the approximation of a single sensel output and consideration of multiple calibration patches. The advantage of the developed method is using local high-pressure zones in calibration patches. The developed method has been successfully applied in calibration of two 5051-350 Tekscan sensors by means of three dead weights: 2 kg, 5 kg and 10 kg. One calibrated sensor was new, and another one had been previously used in the harsh environment of the ice tank in the experiment with model ice. The calibration curves for these two sensors did not reveal a significant difference. For 72% of the 150 obtained load patches in calibration, the absolute discrepancy of actual and calibrated load occurred to be less than 5%.

## 1. Introduction

An interpretation of experimentally measured local loads and pressures is impossible without correct calibration of measuring equipment. Despite the fact that some pressure tools are already widely used in various applications, discussions on the methods and effectiveness of their calibration still take place. This is because pressure measurements are carried out by groups of researchers from different fields who operate in very different technical conditions. Thin tactile pressure sensors based on the use of piezoresistive ink can be the example of such a measuring tool. The principles of manufacturing such pressure sensors were developed about 40 years ago at the Massachusetts Institute of Technology [1]. Later, such sensors and the equipment for their operation have been manufactured by Tekscan (USA).

Tekscan tactile pressure sensors are glued polyester sheets with piezoresistive ink strips applied to them. The intersections of the strips form sensitive elements—so called, sensels. The 8-bit signals received from these elements when the sensor surface is loaded give the resulting pressure matrix measured by the sensor. For protection purposes, the sensor is additionally laminated. The terminal of a sensor sheet with TAB-area connects with the Handle, which contains Amplifiers, Multiplexers, Control and other blocks, and can support the sampling rate up to 1000 Hz [2]. However, in the 7.70-16I version of the I-Scan Versatek System, which is used by the author, the maximum sampling rate is indicated at 730 Hz.

The sensors manufactured by Tekscan have been used in the studies on the pressures in the soil [3,4,5,6,7], railroad track pressure [8], pressure in the joints [9,10], pressure by medical pressure garments and socks [11,12,13], pressure when gripping tools and handles [14,15,16], pressures between rolling elements [17] and others. This variability of applications is provided by the very small thickness of the sensors (usually 0.1–0.3 mm), their flexibility and relative unpretentiousness to external conditions. In addition, the sensors are available in various shapes and density of sensitive elements.

In the software I-Scan, which is supplied by the manufacturer, single-point and two-point (Figure 1a) calibration options are available for tactile pressure sensors [18]. Multipoint (Figure 1b) calibration is limited by no more than 10 measurements. I-Scan apparently uses the averaging of the measured pressures over the patch area for calibration, which requires the flatness of both the base material and the indenter exerting the load. For example, the work [19] indicates that a servohydraulic materials-testing machine (Model 8500, Instron Corp., Canton, MA, USA) has been used for calibration. Despite the conducted balancing procedures and fine grounding of the metal pad, the pressure patch on this pad provided in the paper [19] demonstrates the wide range of various pressure values. They vary from small to very high, with the highest pressure zone occupying only 8.4% of the overall patch area.

Accuracy and repeatability [9,20] are considered as key factors in calibration, which ensure successful investigations are performed by means of tactile pressure sensors. The calibration options provided in the proprietary software are widely discussed in research papers, including the aforementioned ones. A number of researchers [8,12] describe the insufficient accuracy of these options, and some [21] even claim that the research goals are not achieved due to the problems with accuracy and repeatability of sensors calibration.

Calibration of tactile pressure sensors is carried out using loading machines [8,9,17,19,20], pneumatic devices [11,12] and dead weights [12,16,21].

A calibration formula requires recalculation of raw 8-bit values measured by single sensels into the required physical units of pressure (Pa, mmHg, psi) or load (N). The manufacturer recommends using an exponential calibration formula [18] with two parameters. Stith, who has performed an extensive practical study of railroad track pressures [8], notes that the calibration curves obtained from low and medium calibration pressures on the same Tekscan sensor differed significantly from the curves obtained from the calibration points with high pressures. In particular, one of the coefficients in the calibration formula increased significantly. The author suggests setting a piecewise calibration curve with different parameters for the ranges of low, medium and high pressures.

The validation of the acquired calibration formula consists in comparing applied and calculated load, which sometimes reveals significant discrepancy. Such discrepancy is associated, among other factors, with the use of a pressure interval within the first 15% of the nominal value of the sensor [20,22]. For example, in [3,5] it is noted that for low pressure values (less than 25 kPa in [3] and less than 50 kPa in [5]), the discrepancy was significant. In [8], where 1200 psi sensors were tested, a similar effect was observed for high pressure values with two-point calibration.

There is a rule of thumb stating that pressures of approximately 20% and 80% of expected maximum experimental pressures should be applied in the calibration process. However, this did not help much in obtaining adequate results in some medical applications, where low pressures are of interest. For instance, in medical compressive textile and garment tests the pressures of the range 10–50 mmHg (1.33–6.67 kPa) should be investigated. Thus, [12] writes that the attempts to calibrate 9801 Tekscan sensors were unsuccessful. In [21] it was noted that with 9833E Tekscan sensor it was impossible to measure the pressures exerted by compression garments due to the impossibility of correct calibration. Nevertheless, in [13] it is reported that the 5027 sensor has provided satisfactory results of measuring such pressures in vivo and in vitro, allowing to find out the ratio of pressures from compression garments, for example, for the calf and for the ankle.

Another widespread application of tactile sensors with challenging calibration is measuring pressures caused by soils and granular matters [3,6,7,22]. In particular, in [3], the loads exerted by soil on a pipe moved by a horizontally directed force have been experimentally studied. Calibration of sensors for such an experiment has been carried out by loading the entire surface of the sensor with a system of protective and distributive shield pads with massive lead plates, which required a crane to work with.

Application of tactile sensors for pressure measurements in soils besides the common calibration problem leads to specific challenges. Large variation in values over the loaded patch is a typical feature of pressures measured in the experiments with granular matters. The study [7] demonstrates the calibration of sensors under loading applied to a sample with a contact surface representing medium sand. The presented images of the pressure distribution over the area show significant inhomogeneity with the presence of spot-like high-pressure zones. With each new loading, the distribution of the displayed pressures changed. The works on pressure measurements in soils and in soil-structure interaction using tactile pressure sensors pay considerable attention to the effects of the signal drift. This is because pressures applied are relatively long-lasting.

A large number of investigations with Tekscan tactile pressure sensors are performed to study local ice loads in physical model conditions. The first among such experiments was performed in the USA [23], considering the interaction of fresh-water ice with a vertical wall. Then, sensors were utilized in the ice tank of Aalto University in Finland for measuring local pressures exerted by model ice on a hull of the icebreaker model [24]. Later, the studies on measuring local loads on a cylindrical structure were performed in the ice tank of HSVA (Hamburg, Germany) [25,26,27]. The researchers from South Korea [28] reported obtaining integral model ice loads on different parts of the hull of the model of an ice-resistant vessel. A recent study performed at Hamburg University of Technology [29] carries out an investigation on tactile sensor measurements of local loads caused by failure of a specially prepared granular ice specimen at different test speeds.

A lot of investigators from various research domains consider calibration of 8-bit matrix piezoresistive tactile sensors challenging [5,8,12,22,30]. Tekscan sensors usually appear in geotechnical and ice studies on local pressures. However, there are many other producers and prototype developers of piezoresistive sensors. For example, prototypic solutions of resistive sensors are widely used in micro-elecromechanical applications. The variety of resistive sensors is used in medical applications and especially in orthopedy, including products of Pressure Profile Systems, Inc., Hawthorne, CA, USA, Medilogic and RSScan. There are studies comparing different sensors workability by means of pressure tests [31,32]. For state-of-the-art description of resistive tactile pressure sensors, the following papers can be addressed [33,34,35,36].

Summarizing problems on 8-bit tactile pressure sensors calibration, it is possible to list the following requirements for the method:It should consider measurements of individual sensels in a calibration patch instead of averaging the pressure on the patch.It should work relatively fast.It should take into account multiple calibration patches, which imitate possible ways of loading in an experiment.

In this paper, the algorithm of tactile pressure sensor calibration, which satisfies the aforementioned conditions, is developed. Unlike previous methods of sensors calibration where a special machinery is required, the proposed method makes sense out of high-pressure zones appearing at an uneven loading. In previous works, calibration required special equipment like bladders and super-flat surfaces to ensure the most even pressure distribution on the sensor. Temperature and humidity conditions appropriate for such equipment may differ from conditions of an experiment with calibrated sensors. This can result in errors of the experimental measurements interpretation. The newly developed method allows to perform calibration actions in wide range of environments. Up to now, the progress in calibration of tactile pressure sensors has been based more on equipment and less on the calibration data interpretation. This paper makes an attempt to shift the focus from hardware to analysis of measurements.

The structure of the paper is as follows. Section 1 is introductive. Section 2 describes the equipment used for data acquisition. In Section 3, the multiple patches method for 8-bit tactile pressure sensors calibration is developed. Section 4 is the discussion. Section 5 describes the results of 5051 Tekscan sensors calibration, both for the new and used sensors. The Section 6 is the conclusion.

## 2. Experimental Set-Up

The method presented in the article is designed for dead weight calibration, but can be applied for other calibration approaches. In this study, three calibrated weights of classes E_2_ [37] have been used: 2, 5 and 10 kg. Implemented metal weights did not have specially flattened basements, so light plexiglass platforms with a stress distributing layer attached at one side have been used as stress distributive pads under them. Multiple loadings of the sensor surface have been performed with variation of pad size and its position on the sensor. The weights and pads are shown in Figure 2. The characteristics of dead weights and platforms are provided in Table 1.

For the case study provided in Section 5, tactile Tekscan 5051 sensors (Figure 3A) have been used with a nominal resolution of 350 psi (2413 kPa). These sensors have a sensitive matrix of 55.9 × 55.9 mm, which consists of 44 rows and 44 columns of intersecting strips that contain resistive ink. This way, each of the sensors consists of 1936 sensitive elements or sensels. The principles of tactile sensors operation are described, for example, in [1]. Each elementary square part of the matrix (sensel) has dimensions of 1.27 × 1.27 mm and includes both the actual area of intersection of the strips with resistive ink and an insensitive contour around it. 

In Figure 3 the equipment utilized for recording local pressures in the calibration process is shown: operator’s workplace with I-Scan software (D), VersaTek Hub (C), Versatek Handle (B) with pressure sensor 5051 plugged into it (A). For recording, the regular sensitivity regime S-29 has been utilized, and the sampling rate frequency 200 Hz has been set.

During calibration operations, obtaining the most uniform distribution of pressures in the loaded patch has not been considered. With the dead weights’ calibration, it is almost impossible to achieve uniform distribution of pressures on the sensor, which lies on the surface of the table. Instead, a technique has been developed for obtaining calibration coefficients using the variety of pressure values registered by different sensels in the loaded area.

Figure 4 demonstrates the examples of pressure patches obtained during calibration with weights of 2, 5 and 10 kg on the platforms with sides of 1, 2 and 3 cm. It can be seen that the peak pressures exceed the periphery pressures of the patch by 5–10 times. It should be noted that similar pressure distributions have been observed in local ice pressure measurements performed in the ice tank (see Section 4). In these experiments, the sensor has been mounted on a vertical wooden wall of a model of an offshore structure, half submerged in water. The after-experiment calibration of this sensor is described in Section 5.

## 3. Method

The relationship between load X on a cell in raw units and load f on this cell in Newtons according to the producer’s guide [18] is governed by the power function
(1)$$f\left(X\right)=A\cdot X^{b}$$

Thus, load $F_{T}$ on the whole patch T divided by the factor is A is the following sum:(2)$$\frac{F_{T}}{A}=\sum_{k}X_{k}^{b}$$
where k is the indexation of loaded cells in patch T. Based on the calibration results, the manufacturer [18] recommends taking such parameters as A and b that minimize the function
(3)$$\sum_{\left\{T\right\}}{\left(A\sum X_{T}^{b}-F_{T}\right)}^{2}$$

It should be noted that each of the patches T at dead weight calibration with the use of plexiglass pads provides an individual combination of different values X (see Figure 4). Thus, finding a single (A,b) as an analytic solution of (3) becomes impossible. A numerical finding of (3) solution is possible, but is computationally demanding. The preliminary averaging of all $X_{k}$ in T is helpful in solving (3) using a pair of calibration patches, but usually does not help to learn much about the range of high pressures (Figure 1).

Let us denote a raw sum of readings of all the loaded cells in the patch T with R:(4)$$R=\sum_{k}X_{k}$$

For evaluation of the parameters A and b we will use an approximation of $X_{k}$ with $Y^{m}$, where
(5)$$Y^{m}=3^{m},$$
and m=1.5. Hence, we substitute an 8-bit value of X, which is practically from 3 to 255, with the value from the set {3,9,27,81,243}. To do so, it is required to separate a segment [3,255] with intermediate boundaries $I_{1}$, $I_{2}$, $I_{3}$, $I_{4}$ in a way, presented in Figure 5. Values of X, which are from $[3,I_{1})$ are substituted with 3; values from $[I_{1},I_{2})$ are substituted with 9; from $[I_{2},I_{3})$–with 27; from $[I_{3},I_{4})$–with 81; from $\left[I_{4},255\right]$–with 243. An example of such an approximation will be provided in the case study section.

Substituting all X in a load patch with corresponding Y, an approximation Q of (4) is required:(6)$$Q=n_{1}Y+n_{2}Y^{2}+n_{3}Y^{3}+n_{4}Y^{4}+n_{5}Y^{5},$$
where $n_{1}$, $n_{2}$, $n_{3}$¸ $n_{4}$, $n_{5}$ are counts of obtained values Y, $Y^{2}$, $Y^{3}$, $Y^{4}$, $Y^{5}$ in the approximation, correspondingly. The success of such an approximation depends on the selected I. Let us introduce the function $d_{T}\left(I\right)$ to measure the goodness of a particular I:(7)$$d_{T}\left(I\right)=\frac{\left|R_{T}-Q_{T}\left(I\right)\right|}{C_{T}},$$
here $C_{T}$ is the number of cells in patch T. Function $d_{T}\left(I\right)$ is measured in raw units and shows how a single cell reading differs from the approximation in average. In the following, we consider a set {T} of patches T, |{T}|=N. Thus, the discrepancy value averaged by all the patches will be as follows:(8)$$d_{\left\{T\right\}}\left(I\right)=\frac{1}{N}\sum_{i=1}^{N}d_{T_{i}}\left(I\right)$$

For finding optimal vector $I^{*}$ of all the possible I with respect to the particular sensor type and loading technique represented by the set {T}, we introduce the following criteria:(9)$$I^{*}:d_{\left\{T\right\}}\left(I^{*}\right)=\underset{I\in \left\{I\right\}}{\min}d_{\left\{T\right\}}\left(I\right).$$

Here, {I} is the domain of all I such as
(10)$$3>I_{1}>9>I_{2}>27>I_{3}>81>I_{4}>243.$$

For every set {T}, we can find the vector $I^{*}$ numerically.

Taking into account that $Q_{T_{i}}\left(I^{*}\right)\approx R_{T_{i}}$, we can consider the following system of the equations, i≠j:(11)$$\{\begin{array}{c} Q_{T_{i}}=\frac{F_{T_{i}}}{A}\\ Q_{T_{j}}=\frac{F_{T_{j}}}{A} \end{array}$$

Knowing $F_{T_{i}}$ and $F_{T_{j}}$ as the corresponding calibration loads, bringing right parts to equal values, subtracting the second equation from the first one and dividing the result by the non-zero Y, we obtain the single algebraic equation
(12)$$c_{1}+c_{2}Y+c_{3}Y^{2}+c_{4}Y^{3}+c_{5}Y^{4}=0.$$

For this equation, a root $Y^{*}>3$ is of interest due to the expected shape of the calibration curve. If such a root exists, then we can take the following $b^{*}$ as estimation of b from (1):(13)$$b^{*}=\log_{3}Y^{*}.$$

The estimation of the parameter A can be found from one of the equations (11) with respect to (5). This way, we come to the pair ${\left(A^{*},b^{*}\right)}_{ij}$, which estimates (A,b) in a computationally fast manner by means of two calibration load patches, $T_{i}$ and $T_{j}$. Note that the desired root $Y^{*}$ of (12) exists not for every (11), but in a large number of practical cases it occurred to be existing.

Processing all the pairs of $T_{i}$ and $T_{j}$, i=1…N, j=1…N, i≠j, we obtain a set $\left\{\left(A^{*},b^{*}\right)\right\}$ each element of which provides the estimation of unknown parameters of (1). It seems that if we find the average values of $A^{*}$ and $b^{*}$, it will give us the desired estimation of (A,b), which is more or less good for calibration points from {T}; however, this is not the case.

The investigation of the plots of $\left\{\left(A^{*},b^{*}\right)\right\}$ derived from calibration of sensors 5051 and 5101 has revealed that these scatterplots follow the specific law (see, for example, Section 5). This law can be described by the exponential curve:(14)$$A=\alpha b^{\beta }$$

Finding α and β by the least square method, we can find such a point $\left(A_{0},b_{0}\right)$, $A_{0}=\alpha b_{0}^{\beta }$, which provides minimum for the function:(15)$$E_{\left(A_{0},b_{0}\right)}=\sum_{T\in \left\{T\right\}}{\left(F_{T}-A_{0}\sum_{k_{T}}X_{k_{T}}^{b_{0}}\right)}^{2}\to min$$

We take the evaluated point $\left(A_{0},b_{0}\right)$ as an estimation for unknown parameters A and b.

The proposed algorithm for finding calibration formula parameters can be written as follows:Obtain a set {T} of the calibration patches T;Find $I^{*}$, which provides minimum to $d_{\left\{T\right\}}\left(I\right)$;For each pair $T_{i}$ and $T_{j}$, i,j=1…N, i≠j, solve the system (11). If the solution $Y^{*}>3$, then evaluate ${\left(A^{*},b^{*}\right)}_{ij}$ using (13) and (11);Using the found set $\left\{\left(A^{*},b^{*}\right)\right\}$, get the estimation $\left(A_{0},b_{0}\right)$ as the solution of (15). The obtained pair estimates the parameters (A,b) in the calibration Formula (1).

## 4. Discussion

The main motivation for developing a new calibration method was the need in the proper interpretation of pressure patches obtained in the ice tank experiment. The experiment has been performed by the Laboratory of Ice Engineering Research Fundamentals (SPbPU, St. Petersburg, Russia) in the ice tank of the Krylov State Research Centre (St. Petersburg, Russia). Tekscan sensor 5051 has been mounted on the vertical wall of the model of an ice resistant marine structure experiencing loads caused by model ice. The experiment has been performed according to the inversed motion principles: ice floe was motionless, while the model of the structure was towed by a trolley. Model ice floe with a granular structure and thickness of ~50 mm has been prepared according to the fine grain technology [38].

Earlier, the study of global loads on the model structure with sloped and vertical walls demonstrated their non-stationarity [39]; thus, the aim of the new experiment was to reveal local loads caused by model ice. For this purpose, special software [40] for a tactile sensors local load measurements analysis has been developed. An example of 5051-350 sensor (Sensor1) ice pressure measurements in raw units is provided in Figure 6. It can be seen that the distribution of the pressures in high pressure zone in the left bottom corner of the frame looks like patches from dead weight calibration provided earlier in Figure 4. It was one of the reasons why the focus has been made on uneven patches in the calibration process.

The process of obtaining calibration points using dead weights is influenced by many factors that lead to variation in the obtained measurements. They include the method of placing the load, the time of fixing the measurement after applying the load, the individual characteristics of the sensor used, the unevenness of the surface where the sensor is located and the surface that transmits the applied force. The use of a large number of calibration patches helps to decrease the influence of individual factors on the result obtained.

Using the degree-of-three approximation for a single sensel raw measurement seems quite convenient. The similar use of the degree-of-two was unsuccessful due to the following circumstances:The minimal non-zero value provided by a Tekscan sensor’ sensel was three;A higher degree of the polynomial (12) hampers obtaining its solution

For now, the reasons why the obtained pairs of the parameters $\left(A^{*},b^{*}\right)$ for the calibration Formula (1) line up along the exponential curve remain unknown to the author. It should be noted that a similar effect is observed when calibrating the sensor 5101, which is twice the size of the 5051 and also has a square matrix of 44 × 44.

## 5. Results

The calibration of two 5051-350 sensors, one of which had been used earlier in the ice tank experiment, and another one as a control, was carried out four times during the month, from 21 March to 18 April 2022 (Table 2). A four-time calibration procedure was carried out in order to estimate the possible magnitude of the sensor capacity drop-in time. The number of measured calibration points was different each time and is given in the third column of Table 2.

After receiving a set {T} of calibration patches, an optimal separation vector $I^{*}$ should be estimated for substituting the values measured by single sensels by their degrees-of-three approximations. As soon as integer vector $I=\left(I_{1},I_{2},I_{3},I_{4}\right)$ satisfies (10), the set {I} of all the possible I is finite. The value of $I^{*}$, which provides minimum to function $d_{\left\{T\right\}}\left(I\right)$, can be simply found by the brute force. Still, the computational efforts can be decreased by considering only those $I_{1},I_{2},I_{3},I_{4}$, which are more or less close to the centers of the intervals shown in Figure 5. The problem of finding $I^{*}$ is easier if in both calibration and experimental measurements only small raw values are registered; in this case, only two or three components of I can be considered instead of four, and (12) can be solved analytically. However, all four components of I should be considered in the problem of finding calibration parameters valid for the whole range of possible raw values.

In Table 2, vectors $I^{*}$ are provided for different sets of the calibration patches. They did not coincide with the midpoints of the intervals, which are 6, 18, 54 and 162, respectively. An example of variation of $d_{\left\{T\right\}}\left(I\right)$ for the particular set of the load patches on the sensor 5051 at fixed $I_{1}=6$ and varying $\left(I_{2},I_{3},I_{4}\right)$ is presented in Figure 7. The Figure demonstrates a rather large subset of I where the value of $d_{\left\{T\right\}}\left(I\right)$ was close to minimal.

The third step of the algorithm is to solve the system (11) for all the possible individual pairs of the calibration patches and try to find the root, which is larger than three. In Figure 8, the chart which demonstrates the processing of 49 calibration patches obtained for Sensor2 on 18 April 2022 is provided. The green and cyan signs show calibration patches pairs for which Equation (12) had a root greater than three, while the raw sum approximation absolute error of each patch was less than 10%. Each of these pairs provided a value $\left(A^{*},b^{*}\right)$. Commutation of calibration patches in the pair provides the same result. The orange signs show the pairs, for which the Equation (12) had the required solution, but the raw sum approximation absolute error was larger or equal to 10% at least for one patch from the pair. Values $\left(A^{*},b^{*}\right)$ from such pairs have not been considered later. The resulting chart has a symmetrical form.

In the set {T}, several calibration patches occurred, which have not provided the solution of (11) satisfying $Y^{*}>3$ in a pair with any or almost any other patch. The rows and columns, which correspond to such patches are marked with grey color. Such patches in this set {T} made up about 12% of the total number of the patches. A larger number of such patches at loading with 2 kg of weight is probably connected with a smaller area of the loaded patch, which was 1 cm^2^–about 3% of the overall sensor area. From Figure 8 we can see that the patches with different applied weights provided more (*A*^*^, *b*^*^) than the same applied weight pairs positioned differently on the surface of the sensor. For example, approximation (5) led to obtaining $\left(A^{*},b^{*}\right)$ in about 34% of the pairs “2 kg–5 kg”, 32% of the pairs “2 kg–10 kg”, 25% of the pairs “5 kg–10 kg”. The effectiveness of the pairs “2 kg–2 kg” was 23%, of the pairs “5 kg–5 kg”–14% and of the pairs “10 kg–10 kg”–only 11%.

As mentioned in the previous section, the obtained pairs $\left(A^{*},b^{*}\right)$ do not seem disordered. Hence, simple averaging does not provide proper results in evaluating the single estimation of unknown (A,b). An example of the distribution of pairs of points $\left(A^{*},b^{*}\right)$ is shown in Figure 9. Here, the upper left point $T_{1}$ with coordinates (1; 0.007) is obtained on the basis of two calibration patches with the same weights (10 kg) and pads (9 cm^2^). The rightmost point on the graph, $T_{2}$, has coordinates (1.463; 0.001), obtained on the basis of calibration patches with 5 and 10 kg on the 9 cm^2^ pad. We estimate the tendency in the scatterplot of points $\left(A^{*},b^{*}\right)$ by a power–law curve (14). In Figure 9, such a curve is shown. Table 2 shows the values $\left(A_{0},b_{0}\right)$ minimizing function (16) for each of the sets of the calibration patches. The found values are taken as final estimations of unknown (A,b).

As the key indicator in a calibration procedure, the manufacturer considers the saturation pressure, which corresponds to the 255 raw sensel value. Pressures exceeding the saturation pressure will not be correctly registered by the sensor. If raw units are related to load in Newtons by means of calibration Formula (1), the saturation pressure is the following:(16)$$p_{s}=\frac{A_{0}255^{b_{0}}}{S_{sensel}}$$
where $S_{sensel}$ is the area of a sensel. In the last column of Table 2, saturation pressures calculated according to this formula are provided. From the obtained values we can make a conclusion that the saturation pressure of a sensor can slightly decrease in time.

The resulting calibration curve for the new 5051-350 Sensor2 evaluated from the calibration procedure dated 18 April 2022, is shown in Figure 10 with a black line. The corresponding curve for Sensor1 evaluated from the calibration procedure on the same day is shown with a green line. Two additional curves are plotted for each of the sensors to provide an idea on scattering of the calibration curves delivered by various pairs of the calibration patches. These additional curves correspond to “terminal” cases of $\left(A^{*},b^{*}\right).$

Finally, the actual load on a calibration patch should be compared with the load calculated for this patch by means of the model (1) with $\left(A,b\right)=\left(A_{0},b_{0}\right)$. For this reason, the corresponding values of the errors have been evaluated. These errors are presented in Figure 11 in the form of combined histograms. One can see that for the majority of the patches the absolute discrepancy did not exceed 5%. However, the discrepancy in a range of 5–10% has also been present. Among 150 investigated patches, three had absolute discrepancies in the range of 10–12%.

## 6. Conclusions

The uniform loading of a tactile sensor to obtain a number of calibration patches with evenly distributed pressure is usually a technically challenging task. This procedure is especially difficult if it is required to obtain evenly distributed high pressures. At the same time, calibration patches with high pressures are necessary for the correct calibration of such high pressures measured subsequently in an experiment. The development of an easy-to-perform calibration method should facilitate the wider use of tactile pressure sensors in various technical applications.

The method of substituting 8-bit measurements of individual sensing elements in a matrix with a three-based exponential function has demonstrated its applicability for evaluating the parameters of the exponential calibration formula. The developed approach has allowed to take into account high pressure zones obtained during calibration of a sensor lying on a hard surface.

The Inevitable variation in conditions of dead weight calibration procedure influences every single result based on a pair of loaded patches. Due to this, the proposed method implements multiple trials. In the case study for the Tekscan 5051 sensor, the scatterplot of the evaluated parameters was finely enough approximated by the power–law curve. In the paper, it is suggested to base on this curve evaluating a single pair of parameters for the calibration formula. The formula, obtained in such a way has demonstrated an error less than 5% for the majority of the calibration patches (more than 72% of the 150 patches). This can be considered as a good result.

This new method contributes to the improvement of the 8-bit pressure sensors dead weight calibration procedure. The proposed procedure is based on multiple trials for the reasons of stability in the results. In particular, this method allows an inexpensive and confident method of comparing sensors integral output before and after an experiment.

## Figures and Tables

**Figure 1 sensors-22-07290-f001:**
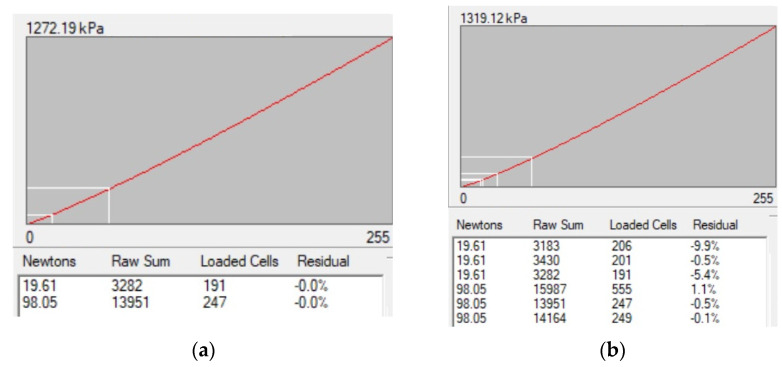
An example of tactile pressure sensor calibration in Tekscan’s I-Scan software: (**a**) 2-points; (**b**) 6-points. The value provided above the plot of the calibration curve is calibrated saturation pressure of the sensor.

**Figure 2 sensors-22-07290-f002:**
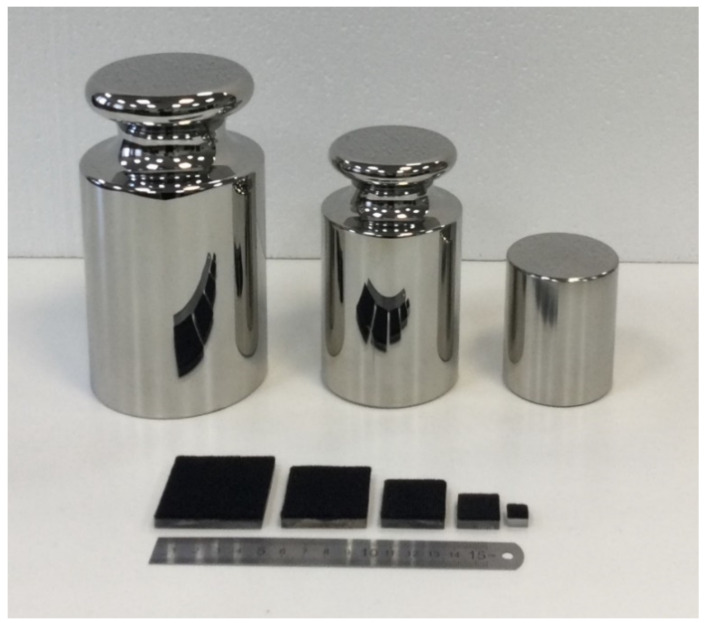
Dead weights (10, 5 and 2 kg, respectively) and pads used in calibration.

**Figure 3 sensors-22-07290-f003:**
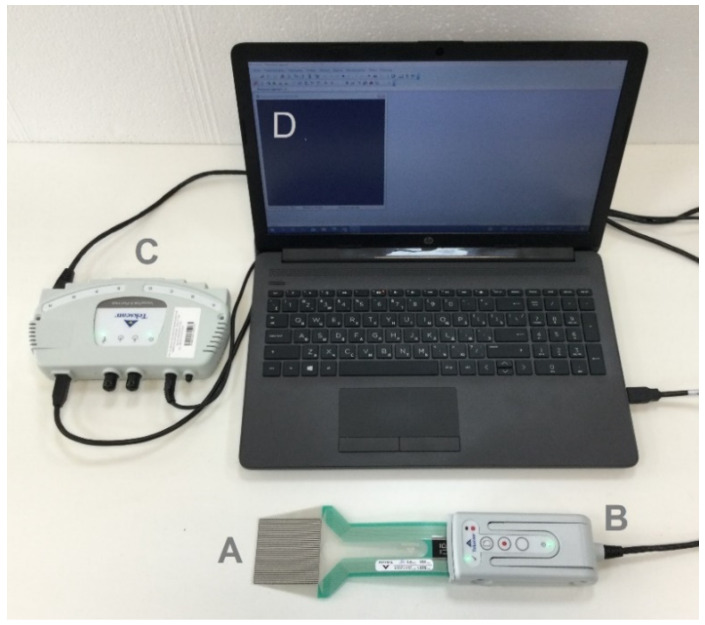
Pressure measuring equipment (Tekscan): (A)—sensor 5051; (B)—VersaTek Handle; (C)—VersaTek Hub; (D) Laptop with I-Scan.

**Figure 4 sensors-22-07290-f004:**
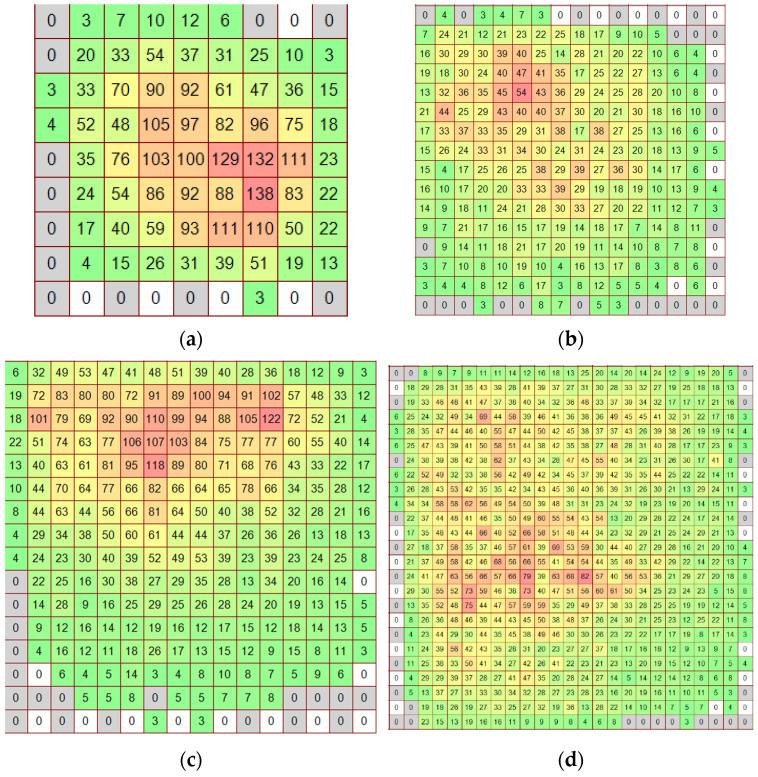
Examples of raw pressure patches at dead weight loading of 5051-350 sensor (regime s29) on various pads: (**a**) 2 kg 1 cm^2^; (**b**) 2 kg 4 cm^2^; (**c**) 5 kg 4 cm^2^; (**d**) 10 kg 9 cm^2^. Green color of cells corresponds to low pressure, yellow—to average pressure, red—to high pressure.

**Figure 5 sensors-22-07290-f005:**
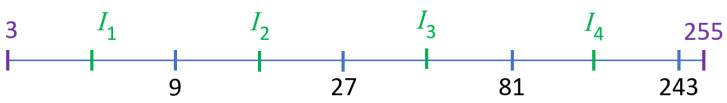
A scheme of the segment [3,255] separation.

**Figure 6 sensors-22-07290-f006:**
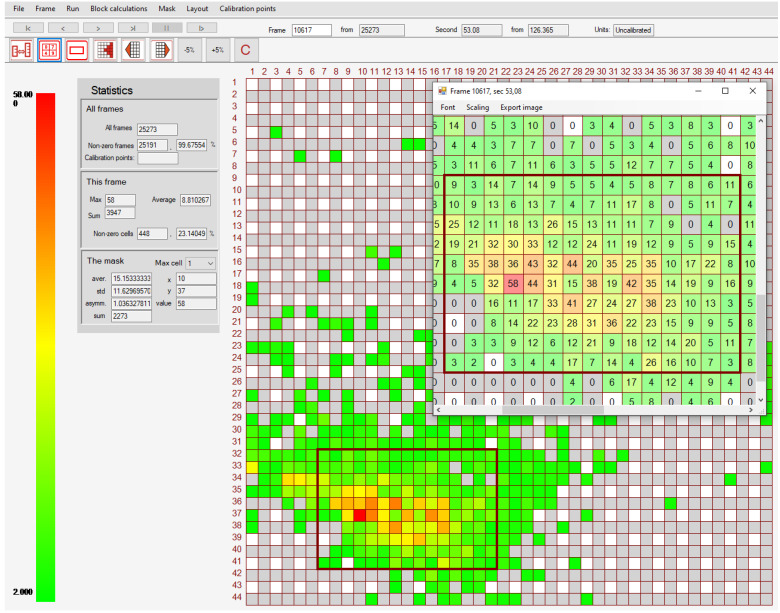
High pressure zone patch on Sensor1 caused by model ice in the ice tank experiment (in raw units). The color legend for raw pressure values is provided in the left part of the figure.

**Figure 7 sensors-22-07290-f007:**
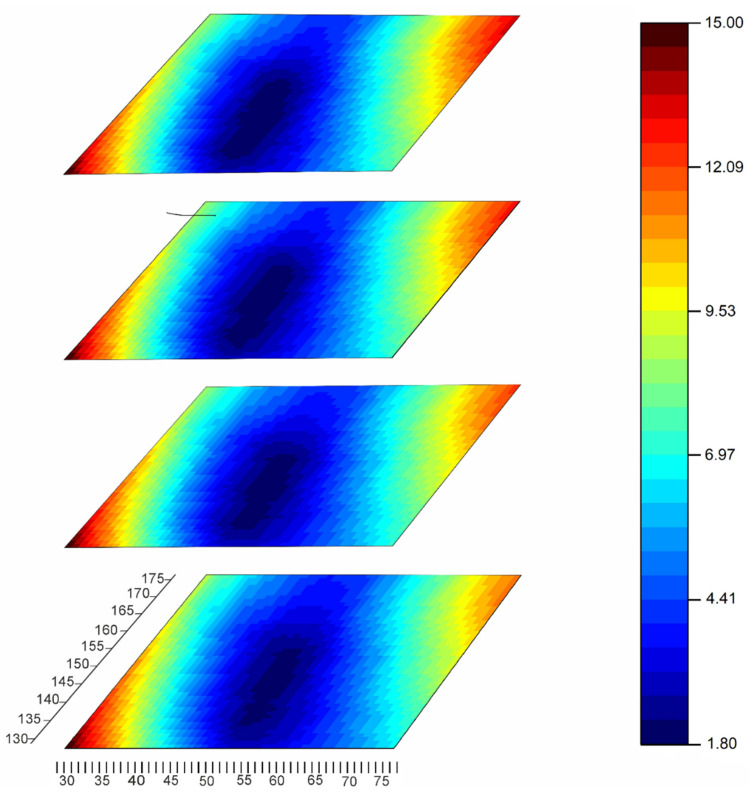
Variation of function $d_{\left\{T\right\}}\left(I\right)$ in calibration of Sensor2 on 18 April 2022, at $I_{1}=6$.

**Figure 8 sensors-22-07290-f008:**
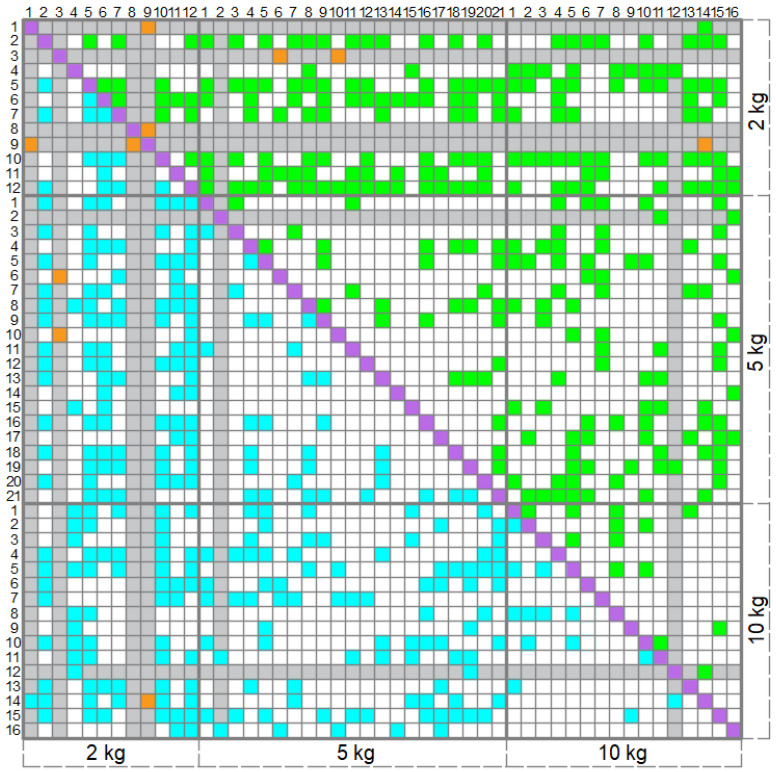
The chart of calibration patches processing, calibration of Sensor2 from 18 April 2022. The explanation on colors is given in the text.

**Figure 9 sensors-22-07290-f009:**
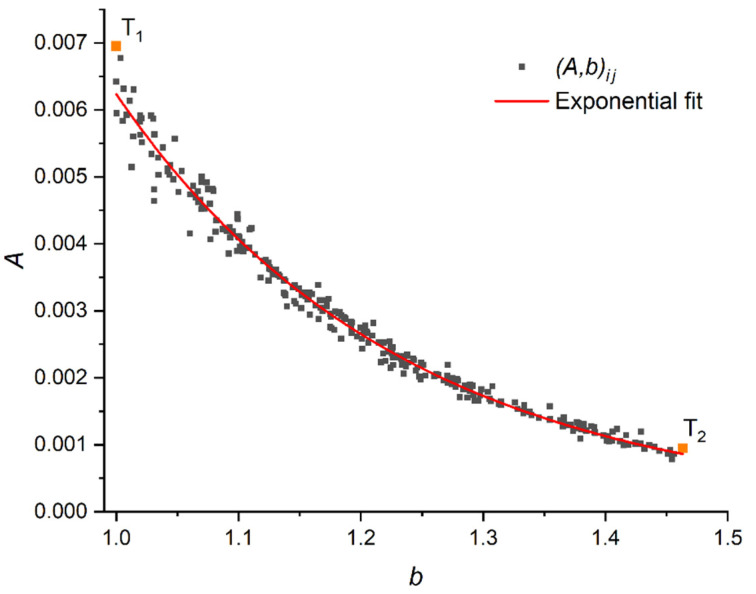
Scatterplot of $\left(A^{*},b^{*}\right)$ for the results of calibration of Sensor2 on 18 April 2022 with the adjusted power–law curve.

**Figure 10 sensors-22-07290-f010:**
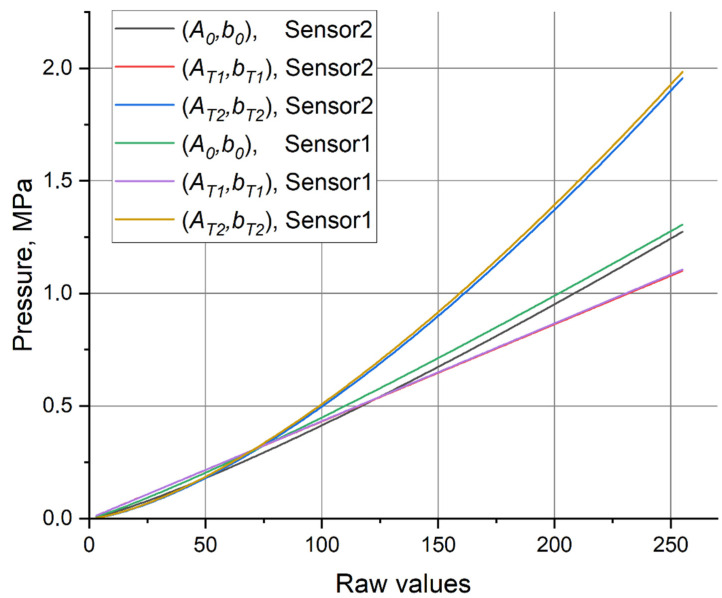
The calibration curves provided by $\left(A_{0},b_{0}\right)$ for Sensor2 (black) and Sensor1 (green) on 18 April 2022 in comparison with the curves provided for these sensors by terminal points $T_{1}$ and $T_{2}$ from the set $\left\{\left(A^{*},b^{*}\right)\right\}$.

**Figure 11 sensors-22-07290-f011:**
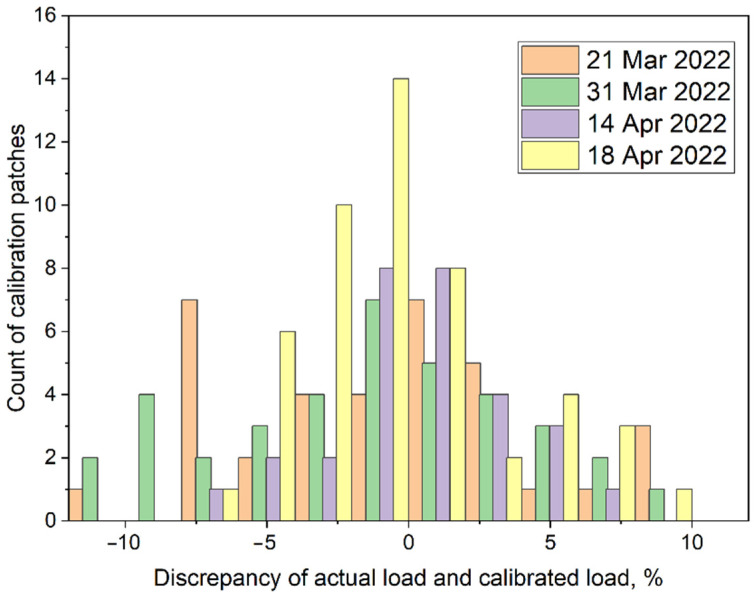
Discrepancy of the applied load on Sensor2 (150 calibration patches) with the load calculated by means of calibration Formula (1) utilizing the parameters $\left(A_{0},b_{0}\right)$.

**Table 1 sensors-22-07290-t001:** Parameters of facilities used in dead weight calibration.

Pad side, cm	1	2	3	4	5
Pad mass, g	0.6	2.6	5.9	10.6	16.1
Pad load, N	0.006	0.026	0.058	0.104	0.158
2 kg with pad load, N	19.619	19.639	19.671	19.717	19.771
5 kg with pad load, N	49.039	49.059	49.091	49.137	49.191
10 kg with pad load, N	98.073	98.093	98.125	98.171	98.225

**Table 2 sensors-22-07290-t002:** Results for the case study.

Sensor	Calibration Date	|{T}|	$I_{1}^{*}$	$I_{2}^{*}$	$I_{3}^{*}$	$I_{4}^{*}$	$d_{\left\{T\right\}}\left(I^{*}\right)$	$A_{0}$	$b_{0}$	$p_{s}$, MPa
Sensor1 (used)	21 March 2022	30	7	17	47	147	1.0	0.00358	1.164	1.403
	31 March 2022	25	6	16	52	137	1.11	0.00398	1.136	1.336
	14 April 2022	23	6	15	51	144	1.17	0.00413	1.120	1.268
	18 April 2022	39	6	15	54	131	1.30	0.00379	1.141	1.306
Sensor2 (new)	21 March 2022	35	6	17	51	144	1.16	0.0023	1.236	1.347
	31 March 2022	37	5	17	48	144	1.03	0.00232	1.227	1.289
	14 April 2022	39	7	17	47	145	1.57	0.00255	1.205	1.252
	18 April 2022	49	99	14	52	147	1.85	0.00265	1.201	1.273

## Data Availability

Not applicable.

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
