# Peer review of "Tactile Pressure Sensors Calibration with the Use of High Pressure Zones"

_sensors, 2022, doi:10.3390/s22197290_

Round 1

Reviewer 1 Report

SUMMARY:

The paper proposed a new method for the calibration of tactile pressure sensors with 8-bit sensels’ output.

SPECIFIC COMMENTS:

Section 1 (Introduction): After the discussion of the state of art, the contribution of the paper is not clear with respect to the other previous works. What is the novelty of your paper?  What are the differences with the other works? Explain how your work fits within the state of the art. (insert this part starting from line {141})

{172}: You have to explain better since it is not clear.

Section 3 (Experimental Equipment and Data Acquisition): Explain what the parameters A and b are. Explain R_T and Q_T(I). In general, try to explain better all the variables and parameters used in the demonstration.

GENERAL COMMENTS:

Grammatical errors and misspellings were founded in many sentences.

The sentences are too long in many parts of the paper, please revise all sentences that exceed 2,5 lines for understandability.

The English proofreading is strictly recommended.

Author Response

Dear Reviewer #1,

thank you very much for your valuable comments. Please find the reply in the attached file.

Reviewer 2 Report

In this manuscript, the authors reported a new method for calibration of tactile pressure sensors with 8-bit sensels output, in which it uses local high pressure zones in calibration patches. And the developed method has been successfully applied in calibration of two 5051-350 Tekscan sensors. Compared with a new calibrated sensor, the other one, which had been previously used in harsh environment of the ice tank in experiment with model ice, did not reveal big differences. From my point of view, the work is generally novel and well organized. Thus, this paper can be received but needs minor revisions before publication. Questions and comments that authors should answer are as follows:

1. In the introduction, the authors did not mention any background of other tactile pressure sensors, and just emphasize Tekscan sensors and software. I do not think it is a good introduction. So the authors should rewrite this part.

Author Response

Dear Reviewer #2,

thank you very much for your valuable comments. Please find the reply in the attached file.

Reviewer 3 Report

The author reported new method for  calibration of tactile pressure sensors with 8-bit sensels' output. This novle metho was very important to tactile pressure sensors calibration. Thus, this manuscript can be accepted after some minor revision.

1.     The form of the manuscript is confused and should be adjusted followed as Introduction, Method, discussion, results and conclusion.

2.  The English and grammar should be improved such as the style of the references and the SPACE etal .

3.     The title of figures 1 should be revised. The right of the inset has "b", but the left of the inset doesn't have "a" . Thus, the titles of the figures should be revised carefully.

Author Response

Dear Reviewer #3,

thank you very much for your valuable comments. Please find the reply in the attached file.
